# Migration, hotspots, and dispersal of HIV infection in Rakai, Uganda

Mary Kate Grabowski[1,2,3] ✉, Justin Lessler[2], Jeremiah Bazaale[3], Dorean Nabukalu[3], Justine Nankinga[3], Betty Nantume[3], Joseph Ssekasanvu[2], Steven J. Reynolds[3,4,5], Robert Ssekubugu[3], Fred Nalugoda[3], Godfrey Kigozi[3], Joseph Kagaayi[3], John S. Santelli[6], Caitlin Kennedy[7], Maria J. Wawer[2,3], David Serwadda[3,8], Larry W. Chang[2,3,5,7] & Ronald H. Gray[2,3]

HIV prevalence varies markedly throughout Africa, and it is often presumed areas of higher HIV prevalence (i.e., hotspots) serve as sources of infection to neighboring areas of lower prevalence. However, the small-scale geography of migration networks and movement of HIV-positive individuals between communities is poorly understood. Here, we use population-based data from ~22,000 persons of known HIV status to characterize migratory patterns and their relationship to HIV among 38 communities in Rakai, Uganda with HIV prevalence ranging from 9 to 43%. We find that migrants moving into hotspots had significantly higher HIV prevalence than migrants moving elsewhere, but out-migration from hotspots was geographically dispersed, contributing minimally to HIV burden in destination locations. Our results challenge the assumption that high prevalence hotspots are drivers of transmission in regional epidemics, instead suggesting that migrants with high HIV prevalence, particularly women, selectively migrate to these areas.

[1] Department of Pathology, Johns Hopkins School of Medicine, Baltimore, MD 21287, USA. [2] Department of Epidemiology, Johns Hopkins Bloomberg School of Public Health, 627 North Washington St., Baltimore, MD 21205, USA. [3] Rakai Health Sciences Program, Old Bukoba Road, P.O. Box 279, Kalisizo, Uganda. [4] Laboratory of Immunoregulation, Division of Intramural Research, National Institute for Allergy and Infectious Diseases, National Institutes of Health, Bethesda, MD, USA. [5] Division of Infectious Diseases, Department of Medicine, Johns Hopkins School of Medicine, Baltimore, MD 21205, USA. [6] Heilbrunn Department of Population and Family Health, Columbia University, 60 Haven Avenue, New York, NY 10032, USA. [7] Department of International Health, Johns Hopkins Bloomberg School of Public Health, 615 N. Wolfe St., Baltimore, MD 21205, USA. [8] Makerere University School of Public Health, Kampala, Uganda. ✉email: mgrabow2@jhu.edu

Human migration facilitates the geographic dispersal of infectious agents at local, national, and global scales, driving epidemics and facilitating pathogen persistence[1,2]. The human immunodeficiency virus (HIV), a leading cause of adult mortality globally[3], is a prime example of the connection between population mobility and infectious disease. Early in the HIV pandemic, migration was found to be a major factor driving dissemination of the virus, and it continues to play a critical role in modern HIV epidemiology[4–6]. A continued understanding of the relationship between migratory dynamics and disease spread is perhaps nowhere more important than in sub-Saharan Africa, where the burden of HIV infections and AIDS-related mortality are concentrated[7,8]. African migrants have higher HIV prevalence[9–15], are less likely to be linked and adhere to antiretroviral therapy (ART), and progress to AIDS more quickly than non-migrants[6,16–18].

Recent data from United Nations Programme on HIV/AIDS (UNAIDS) shows a declining epidemic in sub-Saharan Africa, yet no country is currently on track to meet the 2030 global targets for reductions in HIV incidence[19]. Barriers to reducing HIV incidence include lower ART coverage among youth, men, and mobile and migratory populations, as observed in recent community randomized trials showing limited impact of immediate ART for HIV prevention on population HIV incidence in Southern and Eastern Africa[20–24]. Despite the continued public health threat of HIV, global development spending on the disease has decreased by 20%, necessitating more efficient use of declining resources[25]. This has prompted calls for targeted HIV prevention, including geographic targeting of resources and interventions to high prevalence places[26,27].

Fine-scale mapping of the African epidemic has revealed substantial and widespread variation in HIV prevalence throughout the African continent with one-third of the HIV-infected population concentrated in <1% of its area[28]. Modeling studies of national and sub-national HIV epidemics on the African continent have found that targeting of high prevalence areas (i.e., hotspots) is an efficient use of public health resources, although these studies were conducted in settings where high prevalence areas corresponded to areas with the largest numbers of people living with HIV[29–31]. It is unknown whether targeting of high prevalence areas with a low density of HIV-positive people relative to the surrounding region would have similar impact. For example, fishing communities situated along Lake Victoria have among the highest HIV prevalence levels in Eastern Africa, but these communities have small population sizes and a lower burden of cases relative to the surrounding inland[32,33]. Early modeling work focusing on highly infectious "core groups" suggests that targeting small numbers of infected persons with elevated sexual contact rates, such as Lake Victoria fishing communities, could abate the broader epidemic[34], though the extent to which geographic hotspots or other high prevalence populations function as sources of transmission to another population depends on the degree of connectivity between them as well as the epidemic dynamics within them[35–37].

HIV can be seeded from hotspots into other communities through two mechanisms: cross-community sexual partnerships or migration of HIV-positive individuals between communities. In previous work conducted in rural Rakai District, Uganda and surrounding areas, we showed that sexual partnerships with people outside a community of residence were especially risky for women and were likely responsible for ~25% of incident HIV infections among non-migrants[38]. However, the extent to which migration from hotspots contributes to viral introduction in the Rakai region has not been previously reported. Here, we assess patterns of migration at the individual and community levels and their association with HIV using longitudinal population-based data from persons aged 15–49 years of known HIV status residing in 38 communities in Rakai between 2011 and 2015. Our surveys included four Lake Victoria fishing communities with extremely high HIV prevalence (~40%)[32]. Specifically, we assess the geographic scale and structure of migration networks linking these communities as well as the contribution of migrant-introduced infection to newly detected HIV cases. We also assess HIV prevalence among migrants moving into and out of fishing community hotspots and lower prevalence populations.

Our results highlight four key findings. First, cross community migration is pervasive in rural Uganda, concentrated among young people, and common in both hotspot and non-hotspot communities. We also find that those who migrate are both more likely to have HIV and less likely be on HIV treatment than those who do not. Third, we find that migrants account for the majority of newly detected HIV cases in our study area. Lastly, our results show that those who move to hotspots come from a more geographically diverse pool of locations and have a higher HIV prevalence than migrants who move elsewhere. Overall, our results suggest that migration is common and associated with untreated HIV infection and that HIV hotspots preferentially attract high-prevalence, geographically diverse populations. These data imply that a deeper understanding of the link between migration and HIV is important for HIV control efforts in sub-Saharan Africa.

## Results

**Cross-community migration is pervasive in rural Uganda**. Rakai District is a predominantly rural district in south-central Uganda (area ~2200 km$^2$, population ~518,000) bordered by Masaka District to the north, Tanzania to the south, and Lake Victoria to the east. The Rakai Community Cohort Study (RCCS) is an open population-based household census and cohort of adults aged 15–49 years conducted in 40 communities in Rakai District and surrounding areas since 1994[39].

To understand the patterns of migrations in Rakai and their role in the spread of HIV, we analyzed data from 38 communities included in two sequential RCCS survey rounds. The first survey, denoted as R15, was conducted between 8 August 2011 and 30 May 2013. The second survey, denoted as R16, was conducted between 21 August 2014 and 30 January 2015. Communities included 27 rural agrarian villages, seven semi-urban trading centers and four Lake Victoria fishing communities. HIV prevalence ranged from 9% to 26% in agrarian communities, 11% to 21% in trading communities, and 38% to 43% in fishing communities.

Of 33,727 unique individuals who were census-eligible for the RCCS (23,415 in R15, 26,084 in R16), 23,633 (70%) were present in the community at time of survey for at least one of the two surveys. Being away for work or school was the most common reason for absence. Refusal rates were low, with 95% ($n = 22,901$) of those eligible and present in the community at time of survey participating (15,880 in R15, 16,851 in R16). Long-term residents —those living in the same community in both study rounds—and in-migrants—those moving into an RCCS community between surveys regardless of origin—participated at similar rates (64% vs. 67%); however, participation rates were lower among out-migrants who moved away from study communities after R15 (59%, Supplementary Table 1).

Of the eligible censused population in R15, 24% ($n = 5,585/23,415$) had out-migrated to another community (almost exclusively outside of the study area), and of the eligible censused population in R16, 21% ($n = 5,498/26,084$) had in-migrated into one of the 38 study communities[32]. The overall rates of out-migration and in-migration estimated from these census data

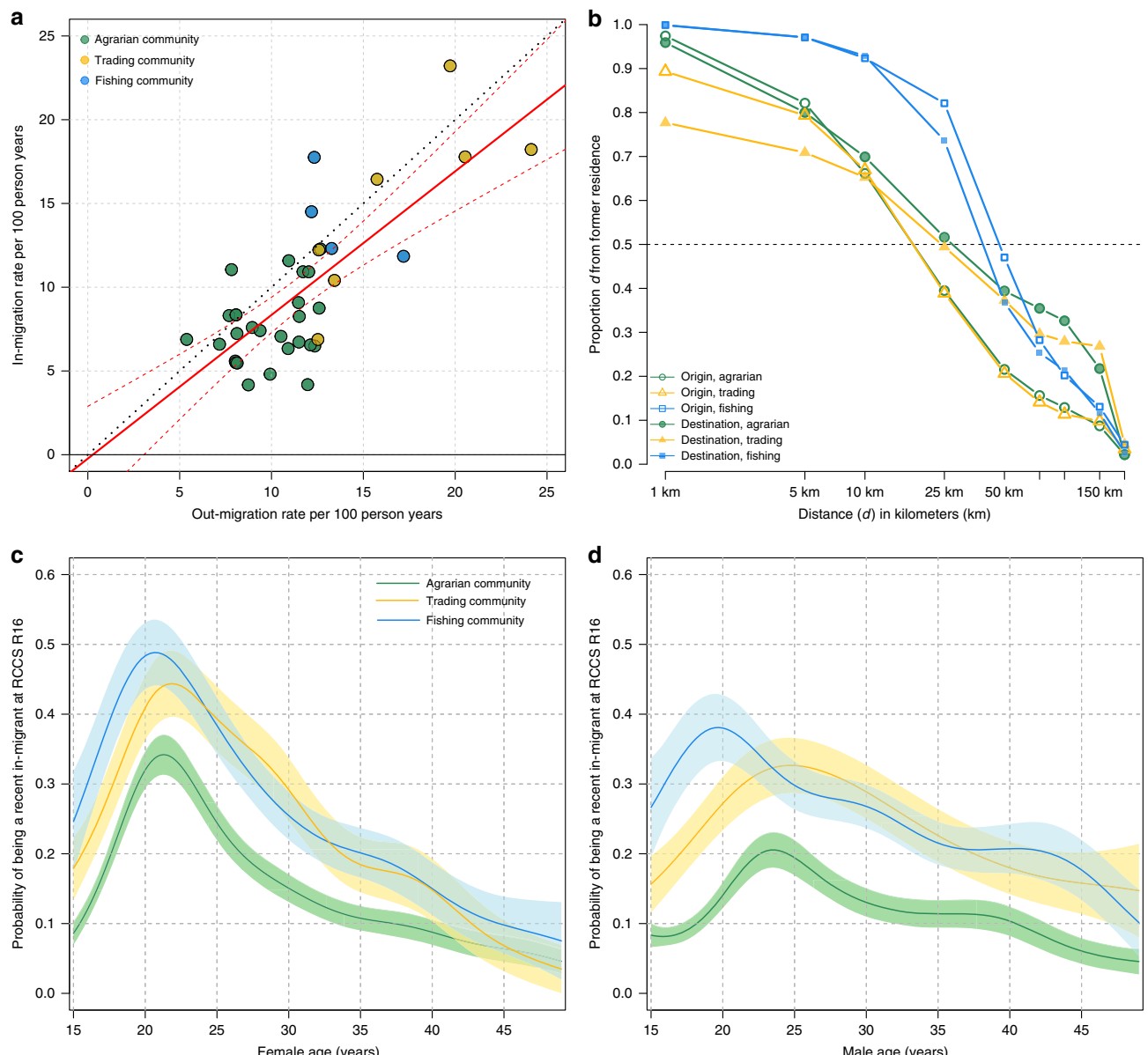

**Fig. 1 Migration dynamics in 38 agrarian, trading, and fishing communities in the Rakai Community Cohort Study. a** In-migration and out-migration rates per 100 person-years in 38 RCCS communities. Agrarian communities are shown in green, trading communities in yellow, and fishing communities in blue. **b** Inverse cumulative distance kernels for place of origin and destination for in-migrants and out-migrants, respectively, showing the proportion who migrated at (or further) particular distances $d$ from origin. Distances are from the source/destination location of each migrant relative to their current/former household in kilometers. **c, d** Proportion of women and men classified as in-migrant at R16 by age and community-type with 95% confidence intervals shown as shaded areas.

were 10.9 and 10.1 per 100 person years (py), respectively. However, migration rates varied markedly across communities (Fig. 1a). Out-migration rates ranged from 5.4 to 24 per 100 py and in-migration rates from 4.2 to 23 per 100 py, with significantly higher levels of in-migration in trading centers (median = 16 per 100 py: IQR: 11–18) and Lake Victoria fishing communities (13 per 100 py, IQR: 12–15) compared to agrarian communities (7.2 per 100 py; IQR: 6.4–8.6; Wilcoxon-rank sum $p$-values < 0.001 for both comparisons). Rates of in- and out-migration were positively correlated at the community-level (coef = 0.86, $R^2$ = 0.55, linear regression $p$-value < 0.001, Fig. 1a).

Among censused eligible persons who participated in at least one of the two surveys ($n$ = 22,901), 29% migrated either into or out of the study communities. Migration was most common among adolescents and young adults aged 15–24 years, with in-migration peaking among men and women in their early 20s (Fig. 1c, d). Similar trends in likelihood of migration by age was observed among out-migrants (Supplementary Fig. 1). Overall, in- and out-migration was more common among women. Among R15 survey participants, 25% of women out-migrated compared to 22% of men (Poisson regression $p$-value < 0.001), and of R16 participants, 24% of women were in-migrants compared to 19% of men (Poisson regression $p$-value < 0.001). In- and out-migrants were less likely to be married and to work in agriculture compared to residents (Table 1). Women most frequently migrated for marriage, for work, or to live with family and friends, while men mainly moved for work or to start a new household.

**Table 1 Demographic characteristics of RCCS participants by migration status and sex.**

| | Women (N = 12,202) | | | | | Men (N = 10,699) | | | | |
|---|---|---|---|---|---|---|---|---|---|---|
| | Long-term residents | Out-migrants[a] | p-value[b] | In-migrants[a] | p-value[b] | Long-term residents | Out-migrants[a] | p-value[b] | In-migrants[a] | p-value[b] |
| Total | 8278 | 1862 | | 2215 | | 7905 | 1422 | | 1450 | |
| Median age (IQR) | 28 (21–35) | 24 (19–30) | <0.001 | 24 (20–30) | <0.001 | 28 (20–36) | 25 (20–32) | <0.001 | 26 (21–33) | <0.001 |
| **Marital status (%)** | | | | | | | | | | |
| Married | 4716 (57) | 990 (53) | | 1349 (61) | | 4172 (53) | 636 (45) | | 708 (49) | |
| Never married | 1969 (24) | 509 (27) | 0.003 | 404 (18) | <0.001 | 2933 (37) | 614 (43) | <0.001 | 570 (39) | 0.012 |
| Previously married | 1593 (19) | 363 (19) | | 462 (21) | | 800 (10) | 172 (12) | | 172 (12) | |
| **Educational status (%)** | | | | | | | | | | |
| None | 1392 (17) | 352 (19) | | 454 (20) | | 1684 (21) | 312 (22) | | 371 (26) | |
| Primary | 3514 (42) | 757 (41) | 0.080 | 882 (40) | <0.001 | 3589 (45) | 609 (43) | 0.187 | 601 (41) | <0.001 |
| Secondary or higher | 3372 (41) | 753 (40) | | 879 (40) | | 2632 (33) | 501 (35) | | 478 (33) | |
| **Primary occupation (%)** | | | | | | | | | | |
| Agricultural/Housework | 3703 (45) | 741 (40) | | 918 (41) | | 1631 (21) | 228 (16) | | 251 (17) | |
| Bar/Restaurant work | 485 (6) | 170 (9) | | 232 (10) | | 27 (0) | 4 (0) | | 9 (1) | |
| Motorcycle taxi/Trucking | 0 (0) | 0 (0) | | 0 (0) | | 146 (2) | 33 (2) | | 51 (4) | |
| Fishing | 6 (0) | 3 (0) | <0.001 | 6 (0) | <0.001 | 1133 (14) | 238 (17) | <0.001 | 312 (22) | <0.001 |
| Student | 1434 (17) | 254 (14) | | 138 (6) | | 1923 (25) | 267 (19) | | 109 (8) | |
| Trader/Shop keeper | 1207 (15) | 294 (16) | | 416 (19) | | 1098 (14) | 201 (14) | | 199 (14) | |
| Other | 1443 (17) | 400 (21) | | 505 (23) | | 1947 (25) | 451 (32) | | 519 (36) | |
| **Place of residence (%)** | | | | | | | | | | |
| Agricultural community | 5571 (67) | 1030 (55) | | 1124 (51) | | 5132 (65) | 753 (53) | | 611 (42) | |
| Trading community | 1158 (14) | 337 (18) | <0.001 | 465 (21) | <0.001 | 842 (11) | 225 (16) | <0.001 | 251 (17) | <0.001 |
| Fish landing site | 1549 (19) | 495 (27) | | 626 (28) | | 1931 (24) | 444 (31) | | 588 (41) | |
| **Reason for move (%)** | | | | | | | | | | |
| Newly married | | 86 (5) | | 588 (27) | | | 0 (0) | | 19 (1) | |
| Divorced/Separated | | 298 (16) | | 5 (0) | | | 44 (3) | | 0 (0) | |
| Work | | 424 (23) | | 571 (26) | | | 614 (43) | | 821 (57) | |
| Started new household | | 115 (6) | | 117 (5) | | | 295 (21) | | 252 (17) | |
| Living with friends/relatives | | 696 (37) | | 815 (37) | | | 196 (14) | | 198 (14) | |
| Don't know/No response | | 232 (12) | | 102 (5) | | | 254 (18) | | 91 (6) | |
| Other | | 11 (0) | | 3 (0) | | | 19 (1) | | 58 (4) | |

[a]There were 231 individuals (women, N = 153; men, N = 78) who migrated from one RCCS study community to another RCCS study community. These individuals are considered both out-migrants and in-migrants.
[b]Chi-square (categorical) and Wilcoxon-rank sum (continuous) p-values; in-migrants and out-migrants are compared to long-term residents only.

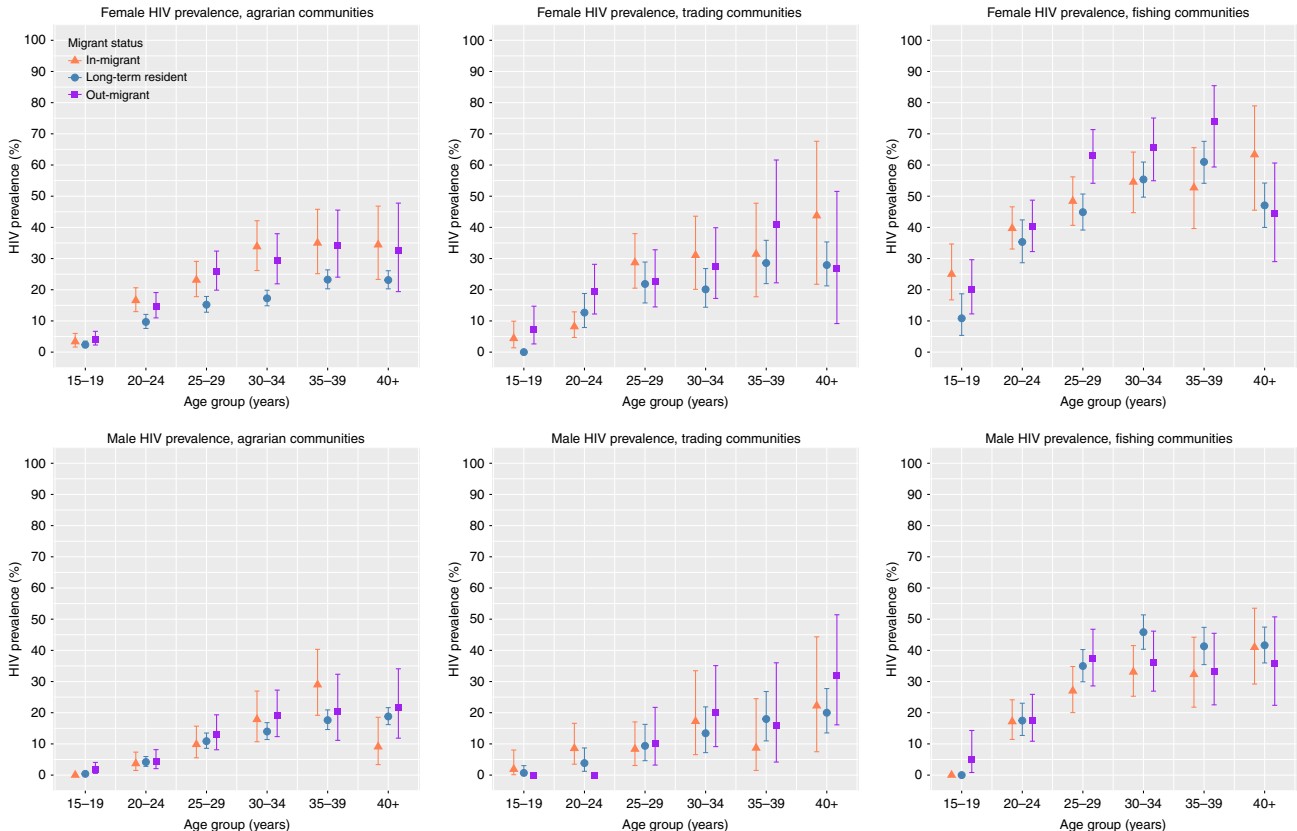

**Fig. 2 HIV prevalence by migration status, sex, and community-type.** Figure shows HIV prevalence with 95% confidence intervals (bars) among in-migrants (orange triangle), out-migrants (purple square), and long-term residents (blue circle) at R16. Prevalence and 95% confidence intervals were estimated using Poisson regression models.

**Hotspots have geographically diverse migrant populations**. We geocoded place of destination for 74% (n = 4,122/5,585) of out-migrants and place of origin for 84% (n = 4,637/5,498) of in-migrants censused, irrespective of survey participation (Supplementary Table 2). Slightly more than half of these in-migrants (56%, n = 2,596/4,637) arrived from communities within Rakai District. A substantial proportion of in-migrants also came from the Masaka District (19%, n = 875), which borders Rakai to the north and Kampala (6%, n = 309), the capital city of Uganda. Of 233 international in-migrants (5% of all in-migrants), 92% (n = 209) arrived from the Kagera District in Tanzania, which is directly south of Rakai. Individuals who out-migrated also tended to move to other Rakai communities (41%; n = 1,792/4,122), Kampala (27%; n = 1,111), Masaka (18%, n = 731), and Kagera (3%, n = 118).

Median distance from place of origin was significantly greater among persons moving into fishing communities (median = 48 km, IQR: 33–84 km) than those moving into either trading (18 km, IQR: 6–44 km, Wilcoxon-rank sum p-value <0.001) or agrarian communities (18 km, IQR: 7–42 km, p < 0.001) (Fig. 1b). Similar trends were observed among out-migrants from fishing communities who moved a median of 44 km (IQR: 24–76) compared to a median of 25 km (IQR: 3–156) and 27 km (IQR: 8–144) among out-migrants from trading and agrarian communities, respectively. Migrants of fishing communities also came from and went to a more geographically diverse set of locations than migrants from other RCCS communities (Supplementary Fig. 2). Out-migrating men and women, younger persons, and HIV-negative individuals traveled further on average than other demographic subgroups or HIV-positive persons (Supplementary Fig. 3).

**HIV prevalence is higher among female migrants**. HIV testing was performed for all consenting RCCS participants during both survey rounds. Among women, age-adjusted HIV prevalence was 30% higher among in-migrants compared to long-term residents across all study communities (Fig. 2, Supplementary Table 3a). Analyses stratified by community-type showed that this disparity was driven by female in-migrants in agrarian communities whose age-adjusted HIV prevalence was 1.64 times higher compared to long-term residents (95% CI: 1.39–1.92). Prevalence of HIV among female in-migrants was somewhat, but not statistically significantly, higher than long-term residents in trading communities (Prevalence Risk Ratio [PRR] = 1.25; 95% CI: 0.95–1.65) and roughly equivalent in fishing communities (PRR = 1.08; 95% CI: 0.93–1.26). HIV prevalence was also significantly higher among female out-migrants compared to long-term residents (PRR = 1.27; 95CI: 1.26–1.58), with the greatest relative difference observed among women in agrarian communities (PRR = 1.50; 95% CI: 1.26–1.58).

HIV prevalence was notably elevated among female migrants in fishing communities, with prevalence among in-migrants peaking at 63% (95% CI: 46–79%) among women over 40 years and at 74% (95% CI: 59–85%) among out-migrant women 35–39 years. Among women 15–19 years, HIV prevalence among in-migrants in fishing communities was 25% (95% CI: 17–35), whereas HIV prevalence among long-term fishing community residents of the same age was 10% (95% CI: 5.4–19%). In contrast, adolescent female prevalence was only 3.4% (95% CI: 1.6–6.0) among in-migrants and 2.4% (95% CI: 1.5–3.5%) among long-term residents in agrarian communities.

While HIV prevalence was generally higher among female migrants compared to long-term residents, HIV prevalence was

**Table 2 Relative risk of out-migration by HIV serostatus, sex, and community type among participants at RCCS R16.**

| Community Type | Probability of out-migration among HIV-negative women (No. out-migrants/Total population) | Probability of out-migration among HIV-positive women (No. out-migrants/Total population) | PRR (95% CI) | p-value | Age adjusted PRR (95% CI) | p-value |
|---|---|---|---|---|---|---|
| **Relative risk of out-migration by HIV serostatus among women** | | | | | | |
| All communities | 21% (1367/6467) | 24% (495/2038) | 1.15 (1.04-1.27) | 0.008 | 1.33 (1.19-1.49) | <0.001 |
| Agrarian communities | 19% (852/4258) | 20% (178/885) | 1.07 (0.91-1.25) | 0.42 | 1.48 (1.25-1.75) | <0.001 |
| Trading communities | 27% (267/987) | 28% (70/246) | 1.05 (0.80-1.36) | 0.71 | 1.25 (0.95-1.62) | 0.11 |
| Fishing communities | 26% (248/952) | 27% (247/907) | 1.05 (0.88-1.25) | 0.62 | 1.22 (1.02-1.47) | 0.032 |
| **Relative risk of out-migration by HIV serostatus among men** | | | | | | |
| All communities | 20% (1199/6102) | 18% (223/1273) | 0.89 (0.77-1.03) | 0.12 | 0.97 (0.83-1.13) | 0.66 |
| Agrarian communities | 17% (681/4028) | 15% (72/486) | 0.88 (0.68-1.11) | 0.29 | 1.16 (0.89-1.48) | 0.24 |
| Trading communities | 27% (203/754) | 23% (22/96) | 0.85 (0.53-1.29) | 0.47 | 1.04 (0.64-1.61) | 0.87 |
| Fishing communities | 24% (315/1320) | 19% (129/691) | 0.78 (0.64-0.96) | 0.019 | 0.84 (0.67-1.03) | 0.10 |

*PRR* prevalence risk ratio, *95% CI* 95% confidence interval, *RCCS* Rakai Community Cohort Study.
ªOverall analysis for all communities adjusted for age and community-type.

not significantly higher among in-migrant or out-migrant men compared to long-term resident men (Supplementary Table 3b). There also were no significant differences between the HIV prevalence of out-migrants and in-migrants of either sex (Supplementary Table 4).

**HIV-positive women migrate more than HIV-negative women.** We also assessed the relative risk of out-migration by HIV serostatus (Table 2). Overall, HIV-positive women were 1.33 times (95% CI: 1.19.49) more likely to migrate compared to HIV-negative women after adjustment for age and community-type, with the greatest differences in out-migration by serostatus observed among agrarian women. In contrast, HIV-positive men were no more likely to migrate than HIV-negative men (Table 2).

**Migrants are the majority of newly detected HIV infections.** There were 1197 persons with HIV infection in R16 who tested HIV seropositive in the RCCS for the first time (Fig. 3a). Of these newly detected cases, 162 (14%) were incident HIV infections (i.e. had a prior HIV-negative test at R15) and 350 (29%) were new cohort enrollees who were long-term residents with unknown duration of HIV infection. The remaining 685 (57%) newly detected cases were in-migrants, 70% ($n = 480$) of whom were women.

In-migrants with newly detected HIV in the RCCS have an unknown duration of HIV infection and may have previously tested positive for HIV. However, HIV-positive in-migrants of both genders were significantly less likely to report ART use than were long-term residents, even accounting for potential biases in survey participation (Table 3, Supplementary Table 5).

Of the 547/685 (80%) in-migrants, who were newly detected HIV cases and for whom a place of origin was known, 281 (51%) originated from within the Rakai District. Of those infections originating outside Rakai ($n = 266$), 51% ($n = 136$) were from Masaka District, 16% ($n = 42$) from Tanzania (all from the neighboring Kagera District), and 8% ($n = 22$) from Kampala (Fig. 3b). HIV-prevalence among all in-migrants from outside Rakai by place of origin ($n = 1138$) is shown in Fig. 3c and was the highest at 28% (95% CI: 20–37%) among those from Tanzania.

**HIV-positive migrants tend to move to hotspots.** We assessed the geography of migratory movements using RCCS data aggregated to the sub-district-level and scaled to reflect local population densities (see Methods). There were nine inland sub-districts

(abbreviated ISD1-9) including agrarian and trading communities and two fishing sub-districts including the fishing communities and four neighboring agrarian communities (abbreviated FSD1-2). The migration network showed strong links between sub-districts containing Rakai's high prevalence fishing communities with Tanzania and Masaka District (Fig. 4a). There were multiple weaker links with Rakai District's inland populations; however, these connections were predominately directed into rather than out of the two fishing sub-districts. Table 4 shows that there were substantially more in-migrants moving from the inland sub-distracts to fishing sub-districts than vice versa, and that the majority of HIV cases among in-migrants were from inland sub-districts. Considering migrant populations from all places of origin and total HIV case burden in the sub-district, we estimated that out-migrants from hotspots contributed to no more than 1.3% (median = 0.4%: IQR: 0–1.1%) of all HIV cases in the nine inland sub-districts.

The reconstructed migration network in Fig. 4a showed that migrant populations with higher HIV prevalence moved into hotspot fishing communities. At the community-level, HIV prevalence among in-migrants was significantly correlated with the HIV prevalence in the long-term residents of destination communities (Fig. 4b). HIV prevalence in most cases was higher among migrants moving into and out of fishing communities regardless of their place of origin or destination (Fig. 4c, Table 4), with larger differentials between migrants and residents observed in women than in men. For example, 45% (95% CI: 34–59%) of female in-migrants from Masaka who in-migrated to fishing communities were HIV-positive compared to 13% (95% CI: 8.0–19%) of women who in-migrated from Masaka to agrarian and trading communities (Poisson regression *p*-value < 0.001). Prevalence of HIV among women who out-migrated to Masaka was also greater among women who originated from one of the fishing communities (56%; 95% CI 41–75%) than from the agrarian and trading communities (19%; 95% CI: 12–29%; Poisson regression *p*-value < 0.001).

Table 4 summarizes HIV prevalence data among long-term residents and in-migrants by place of origin across the nine inland and two fishing sub-districts depicted in Fig. 4a. This analysis was restricted to in-migrants who moved from one of the nine inland or two fishing sub-districts only. The HIV prevalence among in-migrants who moved to an inland sub-district from one of the two fishing sub-districts was somewhat higher compared to the HIV prevalence of in-migrants who originated from one of the nine inland sub-districts (16.1% vs.

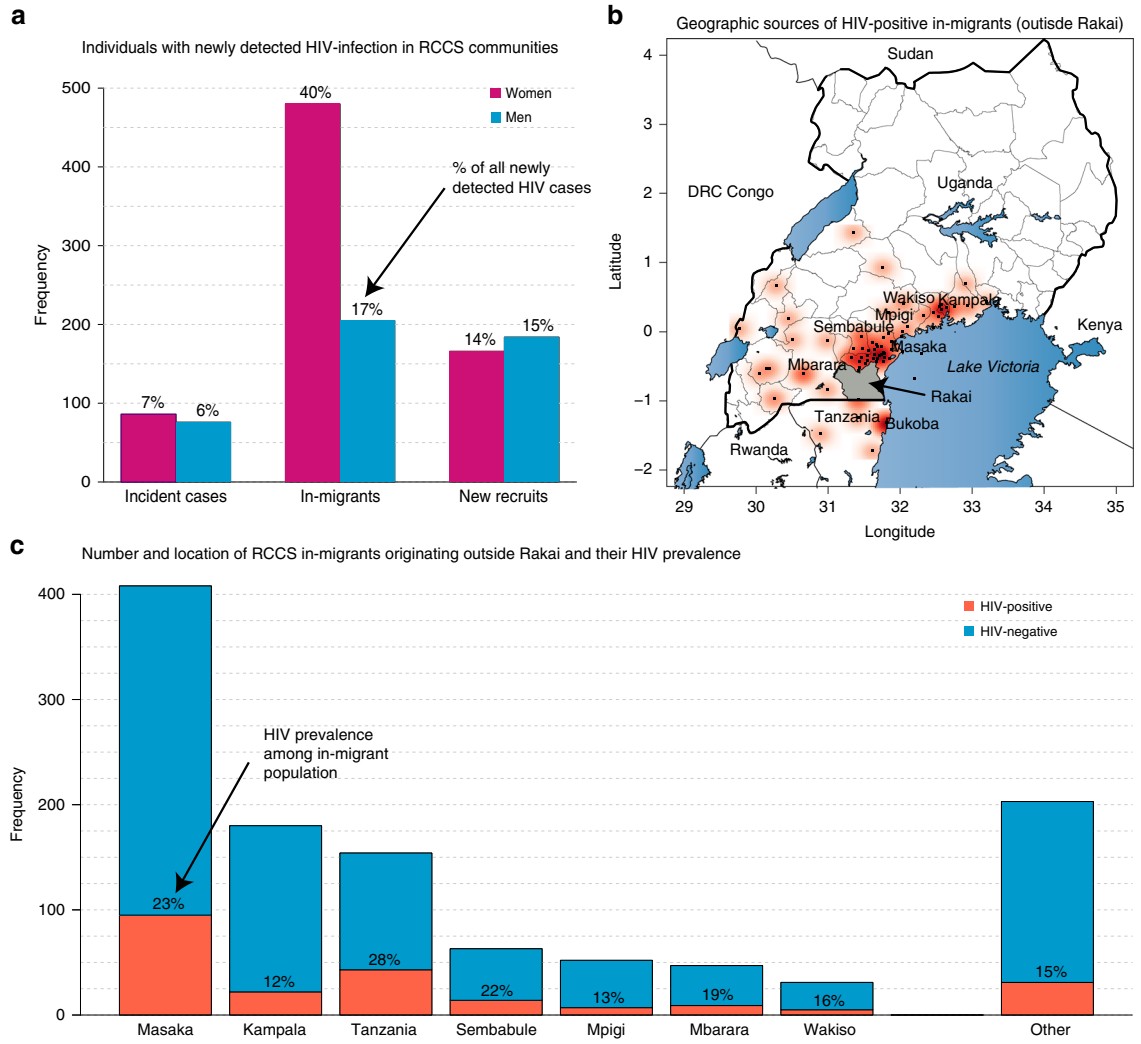

**Fig. 3 Sources of newly detected HIV infections in the Rakai Community Cohort Study.** Newly detected cases were defined as individuals testing HIV seropositive for the first in the RCCS. **a** Frequency and proportion of newly detected incident cases among long-term residents, newly detected cases of unknown duration among in-migrants, and newly detected cases of unknown duration among recently recruited long-term residents. **b** Point density map showing place of origin among HIV-positive in-migrants with darker red areas indicating a higher frequency of in-migrants. **c** HIV prevalence among in-migrating populations originating from districts outside of Rakai.

21.5%), and this difference was driven exclusively by female in-migrants. In comparison, the HIV prevalence among in-migrants who moved to a fishing sub-district was substantially higher than the HIV prevalence among in-migrants who moved to an inland sub-district. HIV prevalence among in-migrants moving to fishing communities did not statistically significantly differ if they originated from an inland or fishing sub-district (35.0% vs. 30.6%).

Since the high prevalence of HIV observed among in-migrant populations in fishing communities could be because individuals rapidly acquire HIV after moving there rather than coming into destination communities already HIV positive, we assessed HIV prevalence by length of stay in fishing communities (Fig. 5). If individuals predominately acquire HIV after rather than before arrival, HIV prevalence should increase with duration of stay. Instead, we found no statistically significant differences in female HIV prevalence by duration of stay. However, HIV prevalence among male in-migrants increased with duration of stay from 18% to 39% over 2 years (Poisson regression $p$-value = 0.006), suggesting a substantial proportion of these male migrants likely acquired infection within the fishing communities.

## Discussion
More than three decades after the first reported AIDS cases in East Africa, migration continues to play an important and complex role in HIV epidemiology[40]. Using detailed data from a population-based cohort, we found that nearly one-third of the Rakai Community Cohort Study's population migrated over a 2-year period. Women were more likely to migrate than men and migrant women were more likely to be HIV-positive than long-term residents. HIV-positive migrants regardless of sex were less likely to use ART than HIV-positive long-term residents and preferentially moved to high-prevalence fishing community hotspots. However, migrants from these same hotspots did not account for a substantial proportion of HIV infections among inland agrarian and trading communities.

Lake Victoria fishing communities have been classified as hotspots because of their extremely high HIV prevalence relative to the general population of Eastern Africa, and in 2013, the Ugandan Ministry of Health recognized fisherfolk as a key population eligible for immediate antiretroviral therapy regardless of CD4 count[41]. One assumption driving this policy was that fishing communities serve as a source of HIV infection to much

**Table 3 Prevalence of self-reported ART use among HIV-positive male and female RCCS participants by migration status, age, and sex.**

| | HIV-positive women (N = 2038), RCCS R15 | | | | HIV-positive women (N = 2034), RCCS R16 | | | |
|---|---|---|---|---|---|---|---|---|
| Age (years) | Long-term residents | Out-migrants | PRR (95% CI) | adjPRR (95% CI) | Long-term residents | In-migrants | PRR (95% CI) | adjPRR (95% CI) |
| 15–24 | 27/217 (12%) | 15/149 (10%) | 0.81 (0.42-1.50) | 0.75 (0.39-1.40) | 77/176 (44%) | 44/188 (23%) | 0.53 (0.37-0.77) | 0.53 (0.36-0.77) |
| 25–35 | 171/730 (23%) | 48/248 (19%) | 0.83 (0.59-1.13) | 0.83 (0.60-1.13) | 348/627 (56%) | 103/271 (38%) | 0.68 (0.55-0.85) | 0.68 (0.54-0.84) |
| 35–49 | 273/596 (46%) | 33/98 (34%) | 0.74 (0.50-1.04) | 0.78 (0.53-1.10) | 462/657 (70%) | 58/115 (50%) | 0.72 (0.54-0.93) | 0.72 (0.54-0.94) |
| All | 471/1543 (31%) | 96/495 (19%) | 0.64 (0.51-0.79) | 0.83 (0.66-1.03) | 887/1460 (61%) | 205/574 (36%) | 0.59 (0.50-0.68) | 0.67 (0.57-0.78) |

| | HIV-positive men (N = 1273), RCCS R15 | | | | HIV-positive men (N = 1244), RCCS R16 | | | |
|---|---|---|---|---|---|---|---|---|
| Age (years) | Long-term residents | Out-migrants | PRR (95% CI) | adjPRR (95% CI) | Long-term residents | In-migrants | PRR (95%CI) | adjPRR (95% CI) |
| 15–24 | 9/97 (9%) | 1/31 (3%) | 0.35 (0.02-1.85) | 0.38 (0.02-2.07) | 8/71 (11%) | 3/36 (8%) | 0.74 (0.16-2.56) | 1.17 (0.24-4.35) |
| 25–35 | 69/480 (14%) | 11/123 (9%) | 0.62 (0.31-1.13) | 0.61 (0.30-1.10) | 145/432 (34%) | 30/117 (26%) | 0.76 (0.51-1.11) | 0.78 (0.51-1.14) |
| 35–49 | 146/473 (31%) | 13/69 (19%) | 0.61 (0.33-1.03) | 0.64 (0.34-1.08) | 279/511 (55%) | 28/77 (36%) | 0.67 (0.44-0.96) | 0.69 (0.45-0.99) |
| All | 224/1050 (21%) | 25/223 (11%) | 0.53 (0.34-0.78) | 0.60 (0.39-0.89) | 432/1014 (43%) | 61/230 (27%) | 0.62 (0.47-0.81) | 0.74 (0.56-0.97) |

*PRR prevalence risk ratio, 95% CI 95% confidence interval, RCCS Rakai Community Cohort Study, adjPRR adjusted PRR; age-stratified analysis adjusted for community-type and overall analysis adjusted for age and community-type.*

larger populations with lower HIV prevalence. However, our analysis of community-based migration networks suggests that HIV transmission dynamics is more complex than the assumption underpinning this geo-targeted initiative, showing that fishing communities predominately received high HIV prevalence in-migrant populations with limited out-bound connectivity to any given inland community. These results are supported by recent phylogenetic analyses from these same study communities and elsewhere in Uganda showing that HIV flows more frequently from inland to Lake Victoria fishing communities than vice versa[42,43].

One hypothesis consistent with our data is that there is a dispersed sub-population of individuals (particularly women) with high HIV prevalence among migrating individuals. This high prevalence population is concentrated in hotspots, such as fishing communities, where they engage in high-risk sexual behaviors and amplify the local HIV epidemic. In Rakai, residents of Lake Victoria fishing communities are predominantly male, more likely to be unmarried, and, irrespective of sex, have substantially higher levels of HIV-related sexual risk behaviors and unprotected sex compared to residents of inland communities[32]. Other research from Rakai and elsewhere in the Lake Victoria basin suggests that female sex work is common in fishing communities and that the mobility of women engaged in sex work is high and transient, so it is possible that this apparent female assortative mixing may be driven by sex work[44–46]. Indeed, a prior qualitative study examining the HIV risk environment within the largest Lake Victoria fishing community in the Rakai study area found that financially vulnerable women working in restaurants and bars had been recruited there by their employers to have sex with fishermen with easily disposable cash incomes[44,47]. Another earlier qualitative study found that unmarried women in Lake Victoria fishing communities had sex almost exclusively with paying partners who were residents within the community[48]. Historically, the role of mobile women in HIV dispersal has received little attention despite higher female than male HIV prevalence and the increasing feminization of migration in Africa[10,49–51]. Additional empirical and model-based transmission studies accounting for these migratory dynamics, population-size, and their possible relationship to sex work could be useful in determining the multifaceted role for high prevalence fishing communities in the broader East African HIV epidemic as well as other hotspots in sub-Saharan Africa.

Overall, migration was a pervasive phenomenon in our study communities with one-third of individuals classified as migrants. Migration was most common among youth in their teens and early twenties, a period during which Africans and women in particular are at high risk for HIV infection[8]. The relatively high mobility in fishing and trading communities is in part due to a seasonal fishing industry and their market economies. A prior study from Rakai showed that migration among youth has been steadily increasing since the 1990s, and that young persons who migrate are significantly more likely to engage in HIV-related risk behaviors[52]. More recent research from Rakai's agrarian and trading populations showed that the first two years following migration is associated with a two-fold higher risk for incident HIV acquisition compared to long-term residents[53]; however, it is unknown whether migrants in hotspots face similar risks. In this study, women moving into hotspots had very high HIV prevalence regardless of the time spent in the fishing communities suggesting that many of these women acquired HIV before they migrated. In contrast, HIV prevalence increased more than two-fold two years after arrival among in-migrant men, implying that these men are at high risk for HIV acquisition after coming into fishing communities. These data suggest a potentially critical role

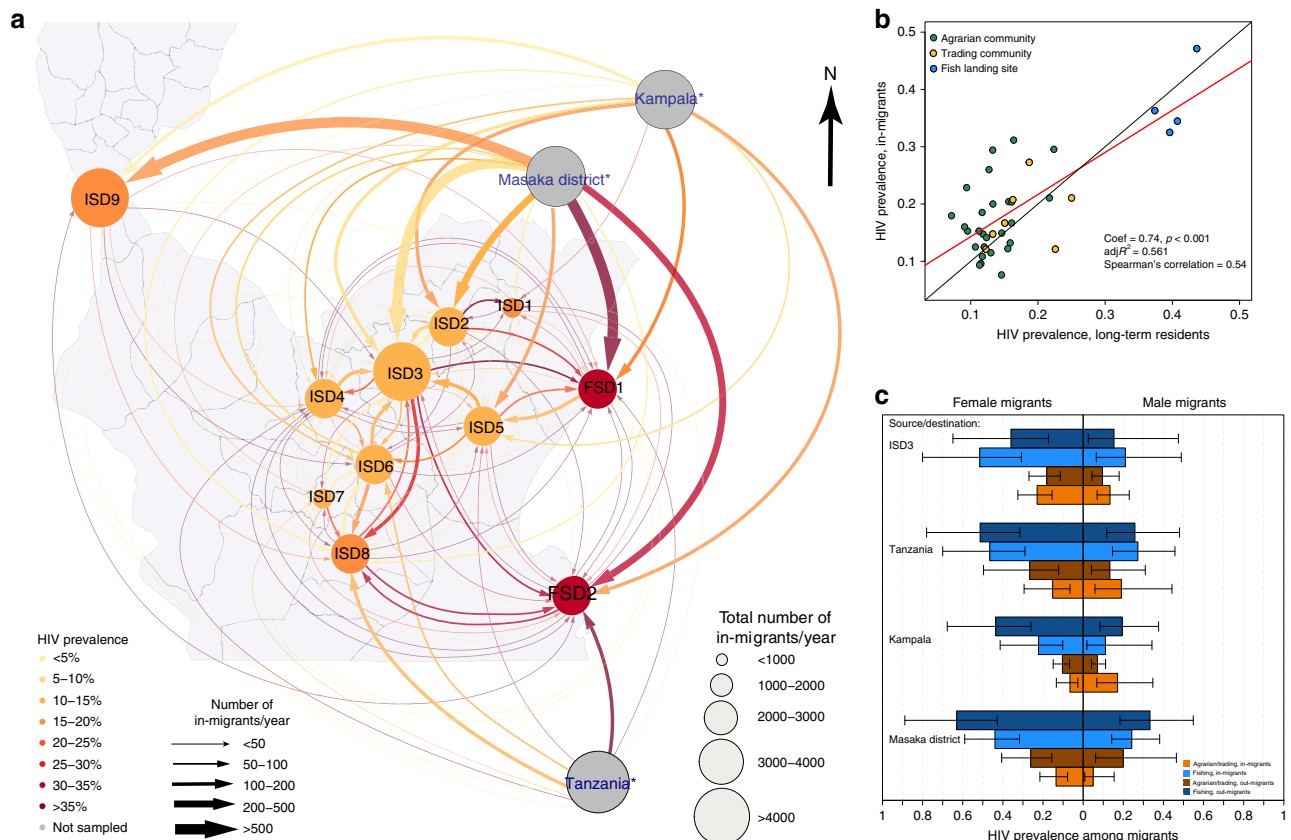

**Fig. 4 Migratory networks and HIV prevalence in the Rakai region. a** Figure shows migration networks at a sub-district level where arrows indicate the frequency of migrants originating from a particular source location. The size of circles corresponds to the size of the total in-migrating population in the sub-district and the size of arrow to the size of the in-migrating population from the associated source location. Color of circles and arrows correspond to HIV prevalence. Labels ISD1–9 denote inland sub-districts 1 through 9 and FSD1–2 fishing sub-districts 1 and 2. Asterisk indicates that the size and color of the circles for Tanzania, Masaka, and Kampala do not reflect the size of migrant populations or prevalence in those locations. **b** HIV prevalence among in-migrants vs. HIV prevalence among long-term residents at the community-level. Agrarian communities are shown in green, trading communities in yellow, and fishing communities in blue. The best fit line was estimated using linear regression and is shown in red. The identify line is shown in black. **c** HIV prevalence among out-migrants from fishing communities (dark blue) and out-migrants from agrarian communities (dark orange) stratified by four places of destination. Also shown is HIV prevalence with 95% confidence intervals (bars) among in-migrants by place of origin and whether they moved into a fish community or a trading/agrarian community (light orange). Prevalence and 95% confidence intervals were estimated using Poisson regression models.

**Table 4 Estimated HIV prevalence among in-migrants in the nine inland (agrarian/trading) sub-districts and the two fishing sub-districts.**

| Destination location | HIV prevalence (95% CI) among long-term residents in destination location | No. of HIV-positive/No. of total in-migrants from inland sub-districts | HIV prevalence (95% CI) among in-migrants from inland sub-districts | No. of HIV-positive/No. of total in-migrants from fishing sub-districts | HIV prevalence (95% CI) among in-migrants from fishing sub-districts | PRR (95% CI) comparing HIV prevalence among in-migrants from fishing vs. inland sub-districts |
|---|---|---|---|---|---|---|
| Women and men | | | | | | |
| Inland sub-districts | 12.9% (12.1%–13.5%) | 140/869 | 16.1% (13.4–18.9%) | 17/79 | 21.5% (12.8–33.4%) | 1.34 (0.79–2.14) |
| Fishing sub-districts | 34.0% (32.1–35.9%) | 90/257 | 35.0% (28.3–42.7%) | 53/173 | 30.6% (23.1–39.6%) | 0.87 (0.62–1.22) |
| Women only | | | | | | |
| Inland sub-districts | 15.4% (14.3–16.5%) | 102/568 | 18.0% (14.7–21.7%) | 17/60 | 28.3% (16.9–44.0%) | 1.58 (0.91–2.56) |
| Fishing sub-districts | 38% (35.6–41.4%) | 64/149 | 43% (33.3–54.4%) | 32/83 | 39.0% (26.7–53.5%) | 0.90 (0.58–1.36) |
| Men only | | | | | | |
| Inland sub-districts | 9.8% (8.9–10.7%) | 38/301 | 12.6% (9.0–17.1%) | 0/19 | 0% | – |
| Fishing sub-districts | 30.0% (27.6–32.5%) | 26/108 | 24.1% (16.0–34.5%) | 21/90 | 23.3% (14.7–34.8%) | 0.97 (0.54–1.72) |

PRR, HIV prevalence estimates, and 95% confidence intervals estimated using Poisson regression; The nine agrarian/trading sub-districts include ISD1–9 (See Fig. 4a for map). The two fishing sub-districts include FSD1–2.
*PRR* prevalence risk ratio, *CI* confidence interval.

for pre-exposure prophylaxis (PrEP) and medical male circumcision among this male sub-population.

There are few population-based studies of internal migration and HIV in Sub-Saharan Africa and, consequently, little is known about the overall contribution of migration to HIV burden at the sub-national level. In the RCCS, recent in-migrants accounted for 57% of all newly detected HIV infections, primarily among female migrants. These data imply that there is a constant introduction of HIV into African communities from external sources. This notion of frequent viral introductions is supported by molecular

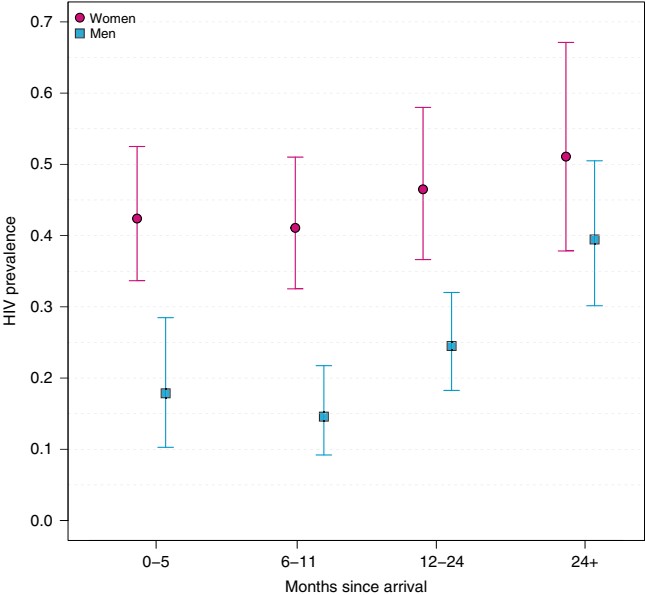

**Fig. 5 HIV prevalence among in-migrants in fishing communities stratified by gender and time since arrival in months.** Female HIV prevalence is shown with circles (pink) and male HIV prevalence with squares (blue). Bars represent 95% confidence intervals. Prevalence and 95% confidence intervals were estimated using Poisson regression models.

epidemiological studies showing multiple circulating strains and higher levels of HIV diversity in main road trading communities within this population[38,54].

We also find lower levels of antiretroviral use among in-migrants irrespective of sex. The relatively low levels of ART use among mobile persons likely results in poorer clinical outcomes and AIDS-related mortality as demonstrated in other settings[6,16–18]. It is unclear whether migrants are a marginalized group with lower levels of ART initiation, or whether migration interrupts treatment use, or both. Though gaps in treatment coverage were greater among in-migrants than out-migrants when compared to residents, both mobile populations had significantly lower levels of ART use after adjustment for age and community type of residence, a finding supportive of the former explanation. There were also substantial changes in HIV treatment guidelines over the study period and scale-up of services may have been sub-optimal among in-migrants from places where the new guidelines were not applicable or were implemented at a slower-pace. The need to deliver HIV care to mobile HIV-positive individuals remains a major public health challenge[6]. Programs that expeditiously identify new in-migrants and link them to care may prove especially useful in these settings.

There are several limitations to this study. Migration dynamics were assessed from census data and do not account for short-term mobility. Notably, the majority of censused individuals who did not participate in the survey were away for work or school. However, a Kenyan study using mobile phone data showed that census data approximate overall patterns of mobility[55]. Participation in the RCCS was also lower among young people, men, and those who out-migrated, but sensitivity analyses adjusting for potential selection bias using inverse probability weights did not significantly change estimates of HIV prevalence or ART use, and participation rates among in-migrants did not substantially differ from those of long-term residents. It is conceivable that participation among out-migrants was lower, because non-participants had already out-migrated prior to the surveys. We also defined migrants as anyone who moved into a study community, irrespective of how far they had moved. Boundaries of study communities were defined by the

RCCS, which may be inaccurate, and even very small movements in residence were classified as migration events. Indeed many of the migration events we observed were hyperlocal, particularly in and around trading communities. We also used self-reported data to assess ART use, which may be subject to reporting bias. However, in a validation study we found a high specificity (99%) and moderate sensitivity (76%) of self-reported ART use compared to detection of antiretroviral drugs in plasma, regardless of migrant status[56]. Lastly, while our results may not be generalizable to hotspots outside of the Lake Victoria basin, they should motivate investigation into the epidemic and population dynamics that give rise to hotspots elsewhere and caution assumptions about them in the absence of data.

In conclusion, we find that migration in rural Uganda is common, particularly among younger persons and women, and that migrants have a higher HIV burden and lower levels of ART use. We also show that high prevalence Lake Victoria fishing communities attract HIV-positive migrant populations. However, migrants from fishing communities do not account for a substantial proportion of HIV infections among adjacent inland communities, casting doubt on the assumption that hotspots necessarily serve as important infectious sources to neighboring general populations. Taken together, our results imply that achieving an AIDS free generation will require special efforts to reach mobile populations.

## Methods

**Data and ethics approval.** We used data from the RCCS, an open population-based census and cohort. Individuals, including migrants, can enter or exit the study population between surveys. The RCCS surveys individuals aged 15–49 in 40 communities in and near the predominantly rural Rakai District of south central Uganda. RCCS communities are classified as agrarian ($n = 27$), trading ($n = 7$), and Lake Victoria fishing communities ($n = 4$). HIV-prevalence is significantly higher in fishing communities (~42%) than in either trading (~14%) or agrarian communities (12%). The study was reviewed and approved by the Ugandan Virus Research Institute's Scientific and Ethics Committee (HS540), the Uganda Council on Science and Technology (GC/127/15/11/137), and Western Institutional Review Board, Olympia WA (20031318). Study participants provided written informed consent at each visit. Antiretroviral therapy and voluntary medical male circumcision were provided by the RHSP and the Ministry of Health through support from the U.S. President's Emergency Plan for AIDS Relief (PEPFAR).

**Identification of classification of migrant populations.** The RCCS conducts a household census with no age truncation prior to the cohort survey. Census data include household GPS location and information on each household member including name, age, gender, marital status, and familial and marital relationships. Individuals who have migrated into or out of a household since the prior census are identified. The RCCS survey, conducted after the census, includes all consenting residents aged 15–49. Interviewers by same sex interviewers use structured questionnaires in the local language (Luganda) to collect sociodemographic, behavioral, and health information. Data are directly entered into mobile PCs and edited in the field.

Migrants were identified at census and defined as persons who moved to or from another community regardless of distance travelled or whether or not the source/destination community was under RCCS surveillance. Specifically, individuals surveyed in R15 who out-migrated to another community prior to R16 were classified as out-migrants. Conversely, individuals who in-migrated into a household from another community between surveys were classified as in-migrants at R16. Persons who did not change communities between the two surveys were classified as long-term residents. To be included in the RCCS survey, in-migrants were required to have stayed in the community for at least 1 month or <1 month but with intention to stay in the study community for six months or longer. In contrast, individuals who resided in the community for less than one month with no intention to stay long-term were classified as visitors. Additional information was obtained on the reason for migration (marriage/divorce, work, living with relatives/friends, other), the movement type (migrated from within or outside a RCCS community), and the community of origin or destination (recorded as a free response). In-migrants provide this information themselves, whereas data on out-migrants was obtained from the head of household or a proxy (i.e., another designated household member) at census. Community of origin and destination for all migrants was identified on a map and geocoded using Google Earth by a team of two Ugandan co-investigators (J.B. and D.N.) with local expertise.

**HIV testing procedures.** HIV testing is performed using a validated three rapid test algorithm[57]. Pre- and post-test counseling and HIV test results are offered by on-site counselors at time of survey. Individuals were considered incident HIV

cases at R16 if they had an HIV-negative test at the prior survey, R15. Newly detected HIV cases included incident HIV cases and newly enrolled individuals testing HIV-positive for the first time in RCCS.

**Statistical analysis**. Demographic characteristics were compared between long-term residents and out- and in-migrants using data from the survey at which they were first observed (R15 or R16). Because HIV prevalence significantly varied between agrarian, trading, and fishing communities, migration dynamics were analyzed separately for these community-types. Out-migration and in-migration rates were defined as the number of migrations per 100 person years among all persons aged 15–49. Cumulative distribution functions, medians and interquartile ranges (IQRs) were used to summarize distances travelled between source/destination locations for migrant populations, and significant differences in travel distance between community types were assessed by Wilcoxon-rank sums tests. Generalized additive binomial models were used to assess the proportions of male and female censused populations who were migrants as a continuous function of age.

To quantify the geographic diversity of migrant populations, we estimated a Shannon entropy score for each community. Shannon entropy is a diversity index that captures the relative proportions and frequencies of different types in a dataset, which in this case was migrants from different geographic locations in a given community[58]. Specifically, Shannon entropy in each community was calculated as $H' = \sum_{i=1}^{R} p_i \ln p_{i}$, where $p_i$ was the proportion of migrants (either in or out) from district $i$. To compare the relative prevalence of ART use among migrating populations and long-term residents, we used Poisson regression with robust variance to estimate prevalence risk ratios (PRR) and corresponding 95% confidence intervals (CI). The sensitivity of our results to differential survey participation by age, gender, and community of residence was assessed using inverse probability weighting as previously described[39,59].

RCCS data were aggregated into sub-districts in order to reconstruct local HIV migration networks for the region. There was a total of eleven sub-districts, including nine inland sub-districts and two fishing sub-districts along the Lake Victoria coast. The total population aged 15–49 years was estimated for each sub-district using World Pop data (population density), RCCS census data on age distributions, and household GPS data using Bayesian hierarchical mapping methods as previously described. RCCS migration data were then extrapolated to these population density maps to obtain the total number of in-migrants in each sub-district as well as in-migrants originating from a given location. A network graph of migrant flows was constructed with a directed edge drawn from sub-district A to sub-district B, if population A accounted for ≥1% of in-migrants per year in population B. Edges and nodes were shaded to reflect to the HIV-prevalence among in-migrating and resident populations, respectively. Masaka District, Kampala, and Tanzania were also included in the network graph since these were identified as key sources of migrant populations; however, only edges directed from these locations were permitted since internal migration data from these populations were not available. Linear regression was used to examine the relationship between HIV-prevalence among in-migrating and long-term resident populations at a community-level. All statistical analyses were performed in the R statistical software (V3.6.1), including network reconstruction, which was done using the igraph package. Maps were created using the raster package and GADM administrative boundaries.

**Reporting summary**. Further information on research design is available in the Nature Research Reporting Summary linked to this article.

## Data availability

The data used to conduct these analyses include personally identifying geographic and health information protected by privacy law and the Institutional Review Boards (IRB) that approved this study. Individual-level data is available upon request to the Rakai Health Sciences Program (www.rhsp.org) and following IRB approval of investigators and analysis protocols. Partial de-identified datasets including basic aggregate-level epidemiological data on key variables (sex, age group, gender, migration status, HIV status, self-reported ART use, and community type) and matrices used to construct network figures are available are available through Github (https://github.com/HopkinsIDD)) and published on Zenodo (https://doi.org/10.5281/zenodo.3533472).

## Code availability

Code to reproduce this manuscript's figures is available through Github (https://github.com/HopkinsIDD) and published on Zenodo (https://doi.org/10.5281/zenodo.3533472).

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

## Acknowledgements

We thank the Rakai Health Sciences Program as well as the participants of the Rakai Community Cohort Study who made this study possible. This study was supported by, the National Institute of Allergy and Infectious Diseases (R01AI110324, U01AI100031, R01AI110324, R01AI102939, K01AI125086-01), the National Institute of Child Health and Development (R01HD070769, R01HD050180), and the National Institute for Allergy and Infectious Diseases Division of Intramural Research, and the Johns Hopkins University Center for AIDS Research (P30AI094189). The funders had no role in study design, data collection and analysis, decision to publish, or preparation of the paper.

## Author contributions

M.K.G., J.L. and R.H.G. designed the study. M.K.G., J.B. and J.S. conducted data analyses. J.S., D.N. and J.B. managed data. B.N., J.N., M.J.W., R.H.G., J.K., L.W.C., C.K., J.S.S., F.N., G.K., D.S. and R.S. oversaw the design of the Rakai Community Cohort Study and census and survey activities. S.J.R. oversaw laboratory activities, including HIV testing. M.K.G., J.L. and R.H.G. wrote the first draft of the paper. All authors assisted in interpretation of analytic findings and reviewed and contributed to writing the final version of the paper.

## Competing interests

The authors declare no competing interests.
