## [Peer Review File · Nature Communications]

Reviewers' Comments:

Reviewer #1:

Remarks to the Author:

This paper focuses on trying to understand the relationship between migration and the HIV epidemic in Rakai, Uganda. This is an important topic and the authors present some interesting data.

The authors state that the results of their study casts doubt on the source-sink hypothesis that they maintain motivates geo-targeted HIV control initiatives focused on Lake Victoria hotspots. Further, they state that their results challenge the hypothesis that HIV hotspots are major sources of HIV infection within the regional epidemic. I disagree with both these statements on the basis of three major comments:

1. The authors assume that the source-sink hypothesis in HIV epidemiology is something that is widely known and agreed upon. I do not think this is the case. The authors should cite studies to justify their assumption.
2. I disagree with the authors understanding of the source-sink hypothesis. They describe the hypothesis as the idea that "geographic areas of higher prevalence serve as sources of infection to areas of lower prevalence". The source-sink hypothesis was first proposed in the field of ecology in 1988, as the authors note, by H. Ronald Pulliam who used a mathematical model to explain the spatial distribution of a species in a meta-population: a source is a patch where reproduction is greater than mortality, and a sink is a patch where mortality is greater than reproduction. In the absence of dispersal from a source to a sink, the population in the sink goes extinct. More recently, the source-sink hypothesis has been used in the field of infectious disease modeling to understanding the maintenance of an epidemic in a meta-population. In this case, a source is a patch where transmission is high enough to be self-sustaining and a sink is a patch where transmission is too low to be self-sustaining. Unless the movement rate of infected individuals from a source to a sink is above a certain threshold level, prevalence in the sink decreases to zero. None of the data that the authors present shows that any of the 38 communities that they are tracking are functioning as sinks. Unless the authors can demonstrate that any of their communities are functioning as sinks they should remove any discussion of source-sink dynamics. It is very possible that all 38 communities that they are studying are sources.
3. I also disagree with the authors statement that the source-sink hypothesis serves as the implicit rationale for several HIV control strategies, including PEPFARs strategy to focus on high prevalence areas for targeted control. The rationale for targeting hotspots is based on cost-effectiveness, which is specified in terms of infections prevented per dollar spent. The higher the incidence rate, the more cost-effective the strategy; incidence is highest in Hotspots. This geographic targeting strategy has been shown to be the most cost-effective strategy case by using mathematical models. These transmission models include geographic heterogeneity in prevalence but do not include mobility, as a consequence none of these models exhibit source-sink dynamics. Consequently, the authors should not claim that the source-sink hypothesis serves as the implicit rationale for targeting Hotspots.

Reviewer #2:

Remarks to the Author:

General

1. Could not review page 15 as the document has been corrupted
2. Also something appears to have gone array with supplement figures 1 and 2

3. Also, in terms of layout, the methods section follows the discussion

4. I find the term "long-term residents" a little misleading – could you amend simply to resident

5. Standardise terms throughout e.g. those referring to residents and fishing communities (e.g. in table 2 state "residents" whereas elsewhere state "long term residents")

6. I struggle with the term "highly HIV-infected migrant populations" (e.g. page 3 and 19) as this could also be taken as high community viral load – preferable to stay with prevalence

7. Figures and tables: please present figures and tables in the same order they are referenced in the text (e.g. fig 1 c-d referenced before fig 1 b) and please ensure all figures and tables are referenced in the text (e.g. table 1, supplement table 1, supplement table 2, and supplement table 3 are referred to in the text – I couldn't see table 2)

8. The discussions should be expanded to support the findings – potential explanations are alluded to but not fully fleshed out – I remain asking questions why fishing communities differ to the other settings, why mobile communities (i.e. they move in and out of areas in relatively similar measures) only impact inwardly, why are there mobile populations (a little more background), why are there the difference we see between men and women (FSW is alluded to but why are men being drawn in – if employment then discuss) – all these questions are addressed to some extent but as a reader I need a little more – perhaps in a fuller summary end paragraph or two

9. In the introduction and/or discussion it would be helpful to briefly highlight the importance of identifying high prevalence areas / populations for clinical / preventative intervention

Specific

1. Abstract: "We find that migrants moving into hotspots (prevalence ~40%) had significantly higher HIV prevalence than migrants moving elsewhere, but that out-migrants from hotspots dispersed broadly, contributing minimally to the HIV burden within individual destination locations." – it would be helpful to explore this conclusion further in your discussion – particularly the point on "contributing minimally" re. out-migration – it would be helpful to further discuss your thoughts on why assortative mixing of in-migrant FSWs and "residents" may be a potential driver in "hotspots" and why this may not be true in the opposite direction (i.e. mobile high prevalence female population out-migrating to elsewhere) – make clear you are speaking proportionally to a resident population rather than at an individual level. Also need to be careful not to make spurious conclusion in relation to a highly stigmatised population (i.e. FSWs)

2. Methods; page 24: "In-migrants provide this information themselves, whereas data on out-migrants was obtained from the head of household or a proxy at census." – please provide further information on this process e.g. what proxy at census?

3. Methods and results (pages 23 and 5): the geographical boundaries of the study area are described twice – please only describe in the methods

4. Methods; page 24: "Migrants were identified at census and defined as persons who moved to or from another community regardless of distance travelled" – the limitation of "regardless of distance travelled" needs to be discussed in full in the limitations section as the boundaries are merely bureaucratic

5.Methods; page 25: "To quantify the geographic diversity of migrant populations at a district-level, we estimated a Shannon entropy score for each community, a diversity measure which captures both the relative proportions and frequencies of migrants from individual locations" – although an old reference is provided, it would be helpful for one or two additional sentences to explain the approach

6.Results; page 5: Everything prior to "Of 33,727 unique individuals who were census..." is either also in the methods or should be in the methods – please amend

7.Results; page 6: "with significantly higher levels of in migration in trading centers (median=16/100py; IQR: 11-18) and Lake Victoria fishing communities (13, IQR:12-15) compared to agrarian communities (7.2/100py; IQR: 6.4-8.6) - beyond presenting IQR it would be helpful to present a summary measure of significance – also /100py missing for fishing communities

8.Results; page 11: "Women who in-migrated were 41% more likely to be HIV positive compared to long-term residents (adjusted prevalence risk ratio [adjPRR]=1.41; 95% CI: 1.27-1.56), and out-migrating women 27% more as likely to be HIV positive (adjPRR=1.27; 95% CI: 1.14-1.41)" – please rephrase to make clear to whom out-migrating women are being compared

9.Results; page 18: "Since individuals may rapidly acquire HIV upon moving into a high prevalence community rather than coming into communities already HIV positive" – be helpful to provide a reference to this assumption and preferably move it to the methods explaining what you did to look at this and then just present results in the results

10.Discussion; page 19: "HIV-positive migrants regardless of gender were less likely to use ART than residents" – please make clear HIV positive residents

11.Discussion; page 19: "A major objective of United States President's Emergency Plan for AIDS Relief is to strategically target geographic areas with a high burden of infection (i.e. hotspots) where resource investment is assumed to have the greatest impact on the epidemic" – could the authors please confirm this is still the case – I ask as discussions at IAS Amsterdam would not, to my ears, suggest this remains a priority

12.Discussion; page 21: sensitivity analyses are mentioned –please describe briefly in your methods

13.Figures: not clear why 1 c-d focus on in-migrants only whereas a and b include both groups – please clarify

14.Figure 2: title difficult to read as right aligned – please amend

15.Figure 2: small point, as you present prevalence as a % in the text would be helpful to do so also in the figure (rather than as a proportion)

16.References: please check and correct (e.g. Tanser et al in twice)

17.Please consider including Tanser et al "Identifying 'corridors of HIV transmission' in a severely affected rural South African population: a case for a shift toward targeted prevention strategies" as this recently published paper would appear to be relevant to your study and conclusions

Reviewer #1

1. This paper focuses on trying to understand the relationship between migration and the HIV epidemic in Rakai, Uganda. This is an important topic and the authors present some interesting data. The authors state that the results of their study casts doubt on the source-sink hypothesis that they maintain motivates geo-targeted HIV control initiatives focused on Lake Victoria hotspots. Further, they state that their results challenge the hypothesis that HIV hotspots are major sources of HIV infection within the regional epidemic. I disagree with both these statements on the basis of three major comments: The authors assume that the source-sink hypothesis in HIV epidemiology is something that is widely known and agreed upon. I do not think this is the case. The authors should cite studies to justify their assumption.

We do not assume that the source-sink hypothesis is something that is widely known or agreed upon in epidemiology and regret if the prior text led the reviewer to believe this to be the case, though we do believe that it is an implicit assumption in some HIV policy. We have revised the abstract and introduction for clarity and to de-emphasize the source-sink hypothesis. We have focused on the importance of migration and its associations with HIV spread in the context of geographic targeting of hotspots and removed reference to source-sink dynamics to avoid confusing the readership.

2. I disagree with the authors understanding of the source-sink hypothesis. They describe the hypothesis as the idea that “geographic areas of higher prevalence serve as sources of infection to areas of lower prevalence”. The source-sink hypothesis was first proposed in the field of ecology in 1988, as the authors note, by H. Ronald Pulliam who used a mathematical model to explain the spatial distribution of a species in a meta-population: a source is a patch where reproduction is greater than mortality, and a sink is a patch where mortality is greater than reproduction. In the absence of dispersal from a source to a sink, the population in the sink goes extinct. More recently, the source-sink hypothesis has been used in the field of infectious disease modeling to understanding the maintenance of an epidemic in a meta-population. In this case, a source is a patch where transmission is high enough to be self-sustaining and a sink is a patch where transmission is too low to be self-sustaining. Unless the movement rate of infected individuals from a source to a sink is above a certain threshold level, prevalence in the sink decreases to zero. None of the data that the authors present shows that any of the 38 communities that they are tracking are functioning as sinks. Unless the authors can demonstrate that any of their communities are functioning as sinks they should remove any discussion of source-sink dynamics. It is very possible that all 38 communities that they are studying are sources.

We thank the reviewer for this helpful discussion. We agree with the meta-population framework and definition of source-sink dynamics provided, and that our simplified description in the prior manuscript did not accurately reflect these nuances. We also agree with the reviewer that there is no data to support the inference that the inland communities are acting as sinks or that the fishing community act as sources of infection (in fact, we disputed this hypothesis). Our data on dispersal patterns of human migration suggest that there are greater numbers of infected persons migrating from inland into fishing communities than vice versa. We also include new data showing that the HIV prevalence of in-migrating and out-migrating population do not differ in fishing communities (see Table 2 in revised manuscript). These data cast doubt on the suggestion that Lake Victoria fishing communities are acting as important sources of HIV transmission to the broader epidemic.

Despite a lack of empirical data to support the role of fishing communities as sources of infection among inland communities, fishing communities have been stigmatized as being sources of infection in the media and the academic literature, and have been prioritized for intervention because of their suspected role as “bridging” the HIV epidemic. Taken together, our results imply that areas of high HIV prevalence cannot be assumed to serve as infection sources to neighboring communities. We have revised the introduction and discussion sections to reflect these points. Critically, we have reframed the manuscript around geographic targeting of hotspots and removed reference to source-sink dynamics in order to avoid confusion.

3. I also disagree with the authors statement that the source-sink hypothesis serves as the implicit rationale for several HIV control strategies, including PEPFARs strategy to focus on high prevalence areas for targeted control. The rationale for targeting hotspots is based on cost-effectiveness, which is specified in terms of infections prevented per dollar spent. The higher the incidence rate, the more cost-effective the strategy; incidence is highest in Hotspots. This geographic targeting strategy has been shown to be the most cost-effective strategy case by using mathematical models. These transmission models include geographic heterogeneity in prevalence but do not include mobility, as a consequence none of these models exhibit source-sink dynamics. Consequently, the authors should not claim that the source-sink hypothesis serves as the implicit rationale for targeting Hotspots.

In a recent high profile mapping study of the African HIV epidemic, Dwyer-Lindren et al. (*Nature*, 2019) found numerous geographic hotspots of very high HIV prevalence throughout the African continent. They concluded that “*Directing treatment and prevention interventions to locations with a relatively small population density, but where HIV prevalence is high....could be more cost-effective and have greater impact compared to always directing resources to high population density areas*”. However, these and similar recommendations carry the implicit assumption that such hotspots serve as sources of transmission to the general population, and that targeting of hotspots may have substantial indirect public health benefit.

The reviewer notes mathematical modeling studies suggest that geographic targeting is the most cost-effective strategy for reducing incidence. We believe the reviewer is referring to a prior publication by Anderson *et al* (*Lancet*, 2014) and an oral abstract at the *CROI* 2018 conference presented by Cuadro *et al*. These studies used data from Kenya and South Africa, respectively, to assess the merits of geographic targeting and found that targeting the highest prevalence area was most effective for reducing HIV incidence. However, in both of these settings the high prevalence areas also had the highest burden in terms of the number of HIV infections. It’s unclear whether targeting smaller, high prevalence communities with lower numbers of HIV infected persons would be more beneficial than targeting lower prevalence, more populous communities. In Uganda, lakeside fishing areas have extremely high prevalence, but relatively small population size, so the burden of infections is not concentrated in these communities, relative to the more populous lower prevalence inland populations (Chang et al. *Lancet HIV*, 2014).

Lastly, we note that areas of high HIV prevalence may not always correlate to areas of high incidence, particularly if infectivity is reduced by high coverage of ART. Furthermore, even if prevalence/incidence is high, hot spot communities may not warrant targeting if the total numbers of infected people are few and connectivity between hotspots and general populations is limited (Azman and Lessler, *Proc Royal*

Soc B, 2015), as may be the case for Lake Victoria fishing communities. We have edited the text to include a discussion of these prior modeling studies and geographic targeting.

Reviewer #2

1. Thank-you for providing me with the opportunity to review this paper. The authors present an interesting study that characterises migratory patterns and their relationship to HIV infection among communities in Rakai, Uganda using population-based data.

The paper is of interest, well written, thoroughly executed, and presents an array of statistical approaches to explore migratory patterns. I suggest a number of amendments / clarifications to take this paper to publication. I also suggest that, if not already done so, a statistician should review the paper.

We thank the reviewer for the positive feedback.

2. Could not review page 15 as the document has been corrupted

We apologize for sending a corrupted document. We have updated the file.

3. Also something appears to have gone array with supplement figures 1 and 2

We have submitted an updated supplemental file. All figures should be intact.

4. Also, in terms of layout, the methods section follows the discussion

We have followed nature communications ordering of sections, which has methods appearing after the discussion section.

5. I find the term “long-term residents” a little misleading – could you amend simply to resident

We would prefer to use the term long-term resident. Migrants are also residents of the communities, only they have arrived more recently. Migrants, by definition in our study, are new residents in the community with intention to stay.

6. Standardise terms throughout e.g. those referring to residents and fishing communities (e.g. in table 2 state “residents” whereas elsewhere state “long term residents”)

We apologize for the confusion. We have standardized the terminology throughout the manuscript.

7. I struggle with the term “highly HIV-infected migrant populations” (e.g. page 3 and 19) as this

could also be taken as high community viral load – preferable to stay with prevalence

We have changed the wording per the reviewer's suggestion.

8. Figures and tables: please present figures and tables in the same order they are referenced in the text (e.g. fig1 c-d referenced before fig 1 b) and please ensure all figures and tables are referenced in the text (e.g. table 1, supplement table 1, supplement table 2, and supplement table 3 are referred to in the text – I couldn't see table 2)

We have ensured that all tables and figures are referenced and in correct order.

9. The discussions should be expanded to support the findings – potential explanations are alluded to but not fully fleshed out – I remain asking questions why fishing communities differ to the other settings, why mobile communities (I.e. they move in and out of areas in relatively similar measures) only impact inwardly, why are there mobile populations (a little more background), why are there the difference we see between men and women (FSW is alluded to but why are men being drawn in – if employment then discuss) – all these questions are addressed to some extent but as a reader I need a little more – perhaps in a fuller summary end paragraph or two

Please see response to comment #11 below.

10. In the introduction and/or discussion it would be helpful to briefly highlight the importance of identifying high prevalence areas / populations for clinical / preventative intervention

We have added the following text to the introduction of manuscript.

“Recent data from United Nations Programme on HIV/AIDS (UNAIDS) shows a declining epidemic in sub-Saharan Africa, yet no country is currently on track to meet the 2030 global targets for reductions in HIV incidence²⁰. Barriers to reducing HIV incidence are hypothesized to include lower ART coverage among youth, men, and “hard to reach” mobile and migratory populations as was observed in recent community randomized trials showing limited impact of immediate ART for HIV prevention on population HIV incidence in Southern and Eastern Africa²¹⁻²⁵. Despite the continued public health threat of HIV, global development spending on the disease has decreased by 20%, necessitating more efficient use of declining resources²⁶. This has prompted calls for targeted HIV prevention, including geographic targeting of resources and interventions to high prevalence places^{27,28}.

Fine-scale mapping of the African epidemic has revealed substantial and widespread variation in HIV prevalence throughout the African continent with one third of the HIV-infected population concentrated in <1% of its area²⁹. Modeling studies of national and sub-national HIV epidemics on the African continent have found that targeting of high prevalence areas (i.e., hotspots) is an efficient use of HIV resources, however these studies were conducted in settings in which the high prevalence areas also happened to correspond to the areas with the largest numbers of people living with HIV^{30,31}. It is unknown whether targeting of high prevalence areas with a low density of HIV-infected people relative to the surrounding region would have similar impact. For example, fishing communities situated along Lake Victoria have among the highest HIV prevalence levels in East Africa, but these communities have small

population sizes and, consequently, a lower burden of cases relative to the surrounding inland. Early modeling work focusing on highly infectious “core groups” suggests that targeting small numbers of infected persons with elevated sexual contact rates, such as Lake Victoria fishing communities, could abate the broader epidemic³², though the extent to which geographic hotspots or other high prevalence populations function as a sources of transmission to another population depends on the degree of connectivity via human mobility between them as well as the epidemic dynamics within them³³⁻³⁵.”

11. Abstract: “We find that migrants moving into hotspots (prevalence~40%) had significantly higher HIV prevalence than migrants moving elsewhere, but that out-migrants from hotspots dispersed broadly, contributing minimally to the HIV burden within individual destination locations.” – it would be helpful to explore this conclusion further in your discussion – particularly the point on “contributing minimally” re. out-migration – it would be helpful to further discuss your thoughts on why assortative mixing of in-migrant FSWs and “residents” may be a potential driver in “hotspots” and why this may not be true in the opposite direction (i.e. mobile high prevalence female population out-migrating to elsewhere) – make clear you are speaking proportionally to a resident population rather than at an individual level. Also need to be careful not to make spurious conclusion in relation to a highly stigmatised population (i.e. FSWs)

We thank the reviewer for this insightful comment. We have added to the discussion of contextual risk factors for HIV in Lake Victoria fishing communities, focusing on why FSW may be driving the observed assortative mixing. We focus on prior quantitative research showing differences in high risk sexual behaviors comparing fishing to inland communities and qualitative findings showing that women working in bars and restaurants were recruited there to have sex with fisherman who have disposable income. We have also cited an earlier qualitative study conducted in 1997 in a Lake Victoria fishing community outside of Entebbe, Uganda which found that 80-100% of sexual contacts of unmarried women were with paying partners who lived in the fishing communities. These studies suggest that Lake Victoria fishing communities are unique in attracting FSW who tend to have sex with local fishermen once they arrive.

“One hypothesis consistent with our data is that there is a dispersed, highly mobile population of individuals (particularly women) with high HIV prevalence. This population is concentrated in hotspots such as fishing communities, engaged in high-risk behaviors and amplifying the local HIV epidemic. In Rakai, residents of Lake Victoria fishing communities are predominately male, more likely to be unmarried, and, irrespective of sex, have substantially higher levels of HIV-related sexual risk behaviors and unprotected sex compared to residents of inland communities³⁸. Other research from Rakai and elsewhere in the Lake Victoria basin suggests that female sex work is common in fishing communities and that the mobility of women engaged in sex work is high and transient, so it is possible that this apparent female assortative mixing may be driven by sex work⁴¹⁻⁴³. Indeed, a prior qualitative study examining the HIV risk environment within Kansensero, the largest Lake Victoria fishing community in the Rakai study area, found that financially vulnerable women working in restaurants and bars had been recruited there by their employers to have sex with fishermen with easily disposable cash incomes^{41,44}. Another earlier qualitative study found that unmarried women in Lake Victoria fishing communities had sex almost exclusively with paying partners who were resident within the community⁴⁵. Historically, the role of mobile women in HIV dispersal has received little attention despite higher female than male HIV prevalence and the increasing feminization of migration in Africa^{10,46-48}. Additional empirical and model-based transmission studies accounting for these migratory dynamics, population-size, and their possible relationship to sex work could be useful in determining the multifaceted role for high prevalence fishing communities in the broader East African HIV epidemic as well as other hotspots in Sub-Saharan Africa.”

11. Methods; page 24: “In-We migrants provide this information themselves, whereas data on out-migrants was obtained from the head of household or a proxy at census.” – please provide further information on this process e.g. what proxy at census?

A proxy is someone else living in the household if the head of household is absent. This person is usually designated by the head of household to speak on their behalf. We have clarified the text to reflect this point.

12. Methods and results (pages 23 and 5): the geographical boundaries of the study area are described twice – please only describe in the methods

We have made the requested adjustment by deleting the duplicative text in the methods.

13. “Migrants were identified at census and defined as persons who moved to or from another community regardless of distance travelled” – the limitation of “regardless of distance travelled” needs to be discussed in full in the limitations section as the boundaries are merely bureaucratic

We agree that what defines a migrant are artificial administrative boundaries. We have noted this point in the limitations section of the discussion as follows:

“We also defined migrants as anyone who moved into a study community, irrespective of how far they had moved. Boundaries of study communities were defined by the RCCS, which may be inaccurate, and even very small movements in residence were classified as migration events. Indeed many of the migration events we observed were hyperlocal, particularly in and around trading communities.”

14. Methods; page 25: “To quantify the geographic diversity of migrant populations at a district-level, we estimated a Shannon entropy score for each community, a diversity measure which captures both the relative proportions and frequencies of migrants from individual locations” – although an old reference is provided, it would be helpful for one or two additional sentences to explain the approach

We expanded upon the definition of Shannon entropy in the methods section as follows.

“To quantify the geographic diversity of migrant populations, we estimated a Shannon entropy score for each community. Shannon entropy is a diversity index that captures the relative proportions and frequencies of different types in a dataset, which in this case was migrants from different geographic locations in a given community⁵⁵. Specifically, Shannon entropy in each community was calculated as $H' = \sum_{i=1}^R p_i \ln p_i$, where p_i was the proportion of migrants (either in or out) from district i .”

15. Results; page 5: Everything prior to “Of 33,727 unique individuals who were census...” is either also in the methods or should be in the methods – please amend

We appreciate the reviewer's attention to detail on these points. We have revised the text to avoid duplicative information.

16. Results; page 6: “with significantly higher levels of in migration in trading centers (median=16/100py; IQR: 11-18) and Lake Victoria fishing communities (13, IQR:12-15) compared to agrarian communities (7.2/100py; IQR: 6.4-8.6) - beyond presenting IQR it would be helpful to present a summary measure of significance – also /100py missing for fishing communities

We have added Wilcoxon rank-sum p-values (both $p < 0.001$) to the text. We have also amended the text, adding “/100 py” after “fishing communities”.

17. Results; page 11: “Women who in-migrated were 41% more likely to be HIV positive compared to long-term residents (adjusted prevalence risk ratio [adjPRR]=1.41; 95% CI: 1.27-1.56), and out-migrating women 27% more as likely to be HIV positive (adjPRR=1.27; 95% CI: 1.14-1.41)” – please rephrase to make clear to whom out-migrating women are being compared

Due to a small coding error identified at review (i.e. a small fraction of individuals had community type erroneously classified among in-migrants and residents at R16), we have revised this entire section. While this error affected some of the numbers in the manuscript, including data in this particular paragraph, it did not change the central conclusions of the paper or any major findings. The comparison groups are more clearly highlighted in the revised text and all numbers have been verified.

18. Results; page 18: “Since individuals may rapidly acquire HIV upon moving into a high prevalence community rather than coming into communities already HIV positive” – be helpful to provide a reference to this assumption and preferably move it to the methods explaining what you did to look at this and then just present results in the results

This was not an assumption per se. We were noting that the high prevalence of HIV observed among in-migrant populations might be due to either rapid acquisition of HIV after arrival, or coming to communities already HIV-infected elsewhere. To test whether this could be true, we examined HIV prevalence as a function of time since arrival. If individuals were acquiring HIV shortly after arrival, the HIV prevalence would increase with duration of stay. While we found this to be the case for men, we did not observe this trend in women. We apologize for the confusion and we have clarified this point in the main text as follows.

“Since the high prevalence of HIV observed among in-migrant populations in fishing communities could be because individuals rapidly acquire HIV after moving rather than coming into destination communities already HIV-positive, we assessed HIV prevalence by length of stay in fishing communities (Figure 5). If individuals predominately acquire HIV after rather than before arrival, HIV prevalence should increase with duration of stay. Instead, we found no statistically significant differences in female HIV prevalence by duration of stay. However, HIV prevalence among male in-migrants did increase with duration of stay from 18% to 39% over two years ($p=0.006$), suggesting that a large proportion of these these male migrants likely acquired infection within the fishing communities.”

19. Discussion; page 19: “HIV-positive migrants regardless of gender were less likely to use ART than residents” – please make clear HIV positive residents

We have made the requested change.

20. Discussion; page 19: “A major objective of United States President’s Emergency Plan for AIDS Relief is to strategically target geographic areas with a high burden of infection (i.e. hotspots) where resource investment is assumed to have the greatest impact on the epidemic” – could the authors please confirm this is still the case – I ask as discussions at IAS Amsterdam would not, to my ears, suggest this remains a priority

It is unclear whether this is still policy but it is frequently discussed at international meetings. We have removed this text for the sake of accuracy.

21. Discussion; page 21: sensitivity analyses are mentioned –please describe briefly in your methods

We have added the following lines to the methods.

To compare the relative prevalence of ART use among migrating populations and long-term residents, we used Poisson regression with robust variance to estimate prevalence risk ratios (PRR) and corresponding 95% confidence intervals (CI). The sensitivity of our results to differential survey participation by age, gender, and community of residence was assessed using inverse probability weighting using methods as previously described.

22. Figures: not clear why 1 c-d focus on in-migrants only whereas a and b include both groups – please clarify

Figure c-d show the probability of in-migration by age. Trends were generally similar for out-migrants (i.e. younger people tend to migrate more) so we chose to place these figures in the supplement in order to streamline the paper (See Supplemental Figure 1). We are amenable to including these figures in the main text at the request of the editor.

22. Figure 2: title difficult to read as right aligned – please amend

We have adjusted the figure title.

23. Figure 2: small point, as you present prevalence as a % in the text would be helpful to do so also in the figure (rather than as a proportion)

We have made the requested change.

24. References: please check and correct (e.g. Tanser et al in twice)

We apologize for the error. We have updated and proof read the references.

25. Please consider including Tanser et al “Identifying ‘corridors of HIV transmission’ in a severely affected rural South African population: a case for a shift toward targeted prevention strategies” as this recently published paper would appear to be relevant to your study and conclusions

We have included the suggested reference.

Reviewers' Comments:

Reviewer #1:

Remarks to the Author:

This is a very interesting paper by Grabowski and colleagues, and they have addressed many of my previous concerns regarding their discussion of source-sink dynamics. I have some additional concerns that I would like addressed/clarified:

1) The authors define migrants as individuals who have moved into, or out of, a community between the two study periods (2011/12 and 2014/15). They find that migration is extremely high: 24% of the study population out-migrated and 21% in-migrated. Clearly the propensity to migrate is a characteristic of the "general population" and predominantly happens when individuals are young and are looking for work or getting married, this is the case in many other sub-Saharan African countries. The authors data show that migrants are ~30% of the general population, therefore migrants are not a "sub-population" of highly mobile individuals (who are likely to be sex workers) as the authors suggest in the discussion. This has extremely important implications for prevention strategies and treatment programs. The design of any strategy needs to focus on who migrates, what type of community they move from and to, and why they migrate. I suggest that the authors substantially revise the discussion.

2) What percentage of individuals have moved more than once? If this is a low proportion, migrants should not be described as highly mobile. Whether, or not, migrants are highly mobile has extremely important implications for prevention strategies and treatment programs.

3) The authors found that migrants moving into Hot Spots had significantly higher HIV prevalence than migrants moving elsewhere. Could this be because individuals living in fishing communities were moving among the fishing communities rather than from the other types of communities? I think a 3 by 3 matrix/table, for each gender, would be very useful; on each axis the community-type would be shown. Entries in the matrix would show the number (and %) of out-migrants from each community-type that were in-migrants to each community-type. This would show the mixing pattern among the three types of communities. The mixing pattern has extremely important implications for prevention strategies and treatment programs.

4) The authors found that the age-adjusted prevalence for recent in-migrants (women) was 30% higher than long-term residents but that this finding was only for agrarian communities. Was this due to young people who had grown up in fishing communities moving to agrarian communities looking for work or getting married?

5) The authors should discuss why agrarian communities are more stable, with respect to migration, than trading centers and fishing villages.

6) The authors state that "Migrants are the predominate source of newly detected HIV infections", this is not correct. I believe that the authors mean to say "Migrants are the majority of newly detected cases".

7) What percentage of migrants migrated with a partner?

8)The authors state that migrants are most likely to work in agriculture, is there seasonal agriculture in this region?

9) Please clarify what kind of network analyses were conducted.

10) The authors state that their “results challenge the assumption that high prevalence Hot spots are drivers of transmission in regional epidemics, instead suggesting that highly HIV-infected migrants, particularly women, selectively migrate to these areas.” The authors should make clear, based upon their data: (i) why individuals are migrating to Hot Spots (and are there gender differences?), and (ii) where, with respect to community-type, they are coming from (and are there gender differences?).

Reviewer #2:

Remarks to the Author:

Dear Editor

Thank-you for providing me with the opportunity to re-review the paper characterising migratory patterns and their relationship to HIV infection among communities in Rakai, Uganda using population-based data.

In my previous review (reviewer #2) I suggested a number of amendments / clarifications to take this paper to publication. As requested, I have reviewed the point by point response letter and the revised manuscript (focusing on amendments arising in relation to my suggestions), and can confirm that I am happy that the points I raised in the previous round of review have been satisfactorily addressed.

All the best, Brian Rice

Reviewer #1 (Remarks to the Author):

This is a very interesting paper by Grabowski and colleagues, and they have addressed many of my previous concerns regarding their discussion of source-sink dynamics. I have some additional concerns that I would like addressed/clarified:

We thank the reviewer for their positive feedback and the poignant comments below. We hope we have addressed their major concerns.

1) The authors define migrants as individuals who have moved into, or out of, a community between the two study periods (2011/12 and 2014/15). They find that migration is extremely high: 24% of the study population out-migrated and 21% in-migrated. Clearly the propensity to migrate is a characteristic of the “general population” and predominantly happens when individuals are young and are looking for work or getting married, this is the case in many other sub-Saharan African countries. The authors data show that migrants are ~30% of the general population, therefore migrants are not a “sub-population” of highly mobile individuals (who are likely to be sex workers) as the authors suggest in the discussion. This has extremely important implications for prevention strategies and treatment programs. The design of any strategy needs to focus on who migrates, what type of community they move from and to, and why they migrate. I suggest that the authors substantially revise the discussion.

We agree with the reviewer that the migration is a common phenomenon, and one of our major objectives in this manuscript was to highlight how pervasive migration is and particularly among those during their younger adult years when they are at highest risk of HIV acquisition. We have rephrased the discussion to emphasize this point.

“Overall, migration was a pervasive phenomenon in our study communities with one-third of individuals classified as migrants.”

When referring to “sub-population”, we are specifically referring to is the group of extremely high prevalence migrant populations moving in and out of fishing communities (particularly women). We have clarified the text in the discussion.

“One hypothesis consistent with our data is that there is a dispersed, sub-population of individuals (particularly women) with high HIV prevalence among migrating individuals. This high prevalence population is concentrated in hotspots such as fishing communities, engaged in high-risk behaviors and amplifying the local HIV epidemic in hotspots.”

2) What percentage of individuals have moved more than once? If this is a low proportion, migrants should not be described as highly mobile. Whether, or not, migrants are highly mobile has extremely important implications for prevention strategies and treatment programs.

In this study, we were not able to capture what proportion of migrants moved more than once. From prior qualitative studies and from the data presented in this manuscript showing high in and out migration rates into fishing communities, we hypothesize that the high prevalence migrant population in fishing

communities are highly mobile. For clarity, we have removed the phrase “highly mobile” from the discussion.

3) The authors found that migrants moving into Hot Spots had significantly higher HIV prevalence than migrants moving elsewhere. Could this be because individuals living in fishing communities were moving among the fishing communities rather than from the other types of communities? I think a 3 by 3 matrix/table, for each gender, would be very useful; on each axis the community-type would be shown. Entries in the matrix would show the number (and %) of out-migrants from each community-type that were in-migrants to each community-type. This would show the mixing pattern among the three types of communities. The mixing pattern has extremely important implications for prevention strategies and treatment programs.

This is not simply because individuals in fishing communities are receiving in-migrants from other fishing communities which have high HIV prevalence. Individuals migrating into fishing communities had high HIV prevalence irrespective of their point of origin. We have included a new table in the manuscript (Table 4) to highlight this point beyond figures 4A and 4C. The table summarizes the RCCS 16 migration data between the nine agrarian and trading sub-districts inland and the two fishing sub-districts on Lake Victoria (Kyamukaaka and Kyebe) that underpins figure 4A. What we find is that in-migrants into fishing sub-districts are more likely to come from the inland sub-districts than the fishing sub-districts and that the prevalences among in-migrants from inland sub-districts is the same if not higher than the prevalence among in-migrants from other fishing communities. Figure 4C illustrates the same point but for four unique locations including Masaka, Tanzania, and Kampala which were the major points of origin for in-migrants outside of the Rakai sub-districts themselves.

We have added the following text to the results section.

“Table 4 shows summarizes HIV prevalence among long-term residents and in-migrants by place of origin across the nine inland and two fishing sub-districts depicted in Figure 4A. This analysis was restricted to in-migrants who moved from one of the nine inland or two fishing sub-districts only. The HIV prevalence of in-migrants who moved to an inland sub-districts from one of the two fishing sub-districts was somewhat higher compared to the HIV prevalence of in-migrants who originated from one of the other nine sub-districts (16.1 vs 21.5%). This difference in prevalence was driven exclusively by female in-migrants. Prevalence of migrant populations from either location was higher than the long-term resident population (12.9%). In comparison, the HIV prevalence among in-migrants who moved to a fishing sub-district was substantially higher than those moving to an inland sub-district. HIV prevalence among in-migrants moving to fishing communities did not significantly differ if they originated from an inland or fishing sub-district (35.0 vs. 30.6%).”

4) The authors found that the age-adjusted prevalence for recent in-migrants (women) was 30% higher than long-term residents but that this finding was only for agrarian communities. Was this due to young people who had grown up in fishing communities moving to agrarian communities looking for work or getting married?

Table 4 shows that HIV prevalence among in-migrating women in agrarian/trading sub-districts from other agrarian/trading districts was higher than the resident population in those sub-districts. While HIV prevalence was even higher among in-migrating women from the fishing sub-districts these women comprised a small percentage of all women in-migrating into those areas.

5) The authors should discuss why agrarian communities are more stable, with respect to migration, than trading centers and fishing villages.

The economies of fishing and trading communities are markedly different than agrarian communities. The nature of work in fishing and trading communities (trade markets; seasonal migration for fishing) are among the many reasons why migration rates are likely higher in these communities. We have added the following text to the discussion.

“The relatively high mobility in fishing communities is in part due to a seasonal fishing industry and in trading communities because of their market economies.”

6) The authors state that “Migrants are the predominate source of newly detected HIV infections”, this is not correct. I believe that the authors mean to say “Migrants are the majority of newly detected cases”.

We have rephrased the section heading as suggested.

7) What percentage of migrants migrated with a partner?

We did not look at linked partnerships in this study. We currently have another manuscript in preparation that focuses on looking at migration among stable, cohabitating couples (the subset of partnerships we are able to link in RCCS). In this study we find that men typically migrate with female partners but that this is not the case for women. We also find higher HIV prevalence among these migrant partnerships. We feel that these analyses are outside the scope of this manuscript and have therefore not included this data in the revised text.

8) The authors state that migrants are most likely to work in agriculture, is there seasonal agriculture in this region?

Farming is typically year round but is done mostly in the wet seasons of which there are two per year. Of note, the most common occupation among women who migrated was agriculture whereas this was not the case for men.

9) Please clarify what kind of network analyses were conducted.

We used data on the geographic origins of in-migrants and their current location or residence to reconstruct a directed network between sub-districts in our study area (first order network analysis). Typically, second order network analyses (e.g. calculation of centrality and clustering measures) are what one would consider network analyses. We did not do these second order assessments in our study. For clarity, we have rephrased the text to say “Network reconstruction”.

10) The authors state that their “results challenge the assumption that high prevalence Hot spots are drivers of transmission in regional epidemics, instead suggesting that highly HIV-infected migrants, particularly women, selectively migrate to these areas.” The authors should make clear, based upon their data: (i) why individuals are migrating to Hot Spots (and are there gender differences?), and (ii) where, with respect to community-type, they are coming from (and are there gender differences?).

With respect to first point, 57% of men who moved into hotspots moved for work related reasons, 17% to start a new household, and 14% to live with a relative or a friend. Less than 2% of men moved for marriage. In contrast, only 26% of women moved for work, 27% for marriage, and 37% to live with friends and relatives. With respect to the second point, we have added Table 4 as described in response to comment #3 (Figure 4A also shows the major sources of migrants into hotspots which include Masaka District, Kampala, and Tanzania).

Reviewers' Comments:

Reviewer #1:

Remarks to the Author:

The authors have addressed all of my concerns.